# Sustainability of a Given Ten-Week Motor Skills Training Program for Children with Developmental Coordination Disorder

**DOI:** 10.3390/sports10110164

**Published:** 2022-10-24

**Authors:** Orifjon Saidmamatov, Ko’palov Sanjarbek, Olga Vasconcelos, Paula Rodrigues

**Affiliations:** 1Motor Control and Learning Laboratory, CIFI2D, Centre of Research, Education, Innovation and Intervention in Sport, Faculty of Sport, University of Porto, 4200-450 Porto, Portugal; 2Theory and Methodology of Physical Culture, Faculty of Physical Culture, Urgench State University, Urgench 220100, Uzbekistan; 3Kinesiolab, Laboratory of Human Movement Analysis, Piaget Institute, 1950-157 Almada, Portugal

**Keywords:** MABC-2, retention test, motor skill, pre-school children

## Abstract

The aim of this research was to determine the sustainability of a given ten-week motor skills training program for children with developmental coordination disorder (DCD). Children with DCD in four kindergartens in the Khorezm region of Uzbekistan were selected to take part in the study. Participants were 24 children between 4 and 6 years old (5.25 ± 0.13 years), assigned to an intervention group (IG: 17 children; 10 boys) and a control group (CG: 7 children; 4 boys). The Movement Assessment Battery for Children (MABC-2) was used to assess motor competence pre-intervention, post-intervention, and retention test measurement after 18 months. The retention test results for the IG indicated a decline in all three motor domains of MABC-2. Despite this, the results that were acquired during the retention test came out to be better than the results that were achieved during the pre-test. At the same time, children who were allocated to the CG maintained their performance. The findings suggest that a preschool-based motor skill training program has the potential to increase motor skills in children with DCD. However, the positive benefits produced by the intervention may decrease over the course of time if the intervention is not continued.

## 1. Introduction

Developmental coordination disorder (DCD) is a deferment in motor skill development, particularly coordination of movements, making it difficult for the child to move and do routines that children do every day [1]. Moreover, the feelings of inadequacy accompanying a lack of motor coordination are typically strengthened through connections with peers [2]. Children with DCD may display decreased motivation to engage in physical activities as well as opportunities to improve motor skills and fitness, as Rivilis et al. [3] formerly found. In addition to their motor impairments, they also display reduced levels of health-related physical activity. The literature has pointed out that children with movement impairments typically participate in less physical activity and physical exercise [4]. If timely treatment is neglected, movement impairment symptoms persist in a large percentage of adults. Nevertheless, for children’s and adolescents’ healthy social and physical development, it is essential to spare time for recreational physical activity [5]. Consequently, experts from a variety of fields, including pediatrics, occupational therapy, physiotherapy, and kinesiology, have paid considerable attention to DCD [6]. A variety of interventions focused on motor skill training programs have been developed by researchers and healthcare providers to overcome DCD. These programs are focused on physical activities and usual motor skills involving experiences directed to the development of fundamental motor competence. These programs are focused on physical activities and the development of fundamental locomotor motor skills (running, jumping, and walking), manipulative skills (aiming and catching), and balance skills (stabilization, rotation, static balance, and dynamic balance). Most researchers believe these activities improve children’s movement quality and motor competence and reduce the difficulties of DCD [5,7,8]. In addition, studies have indicated that even short-term motor skill interventions can enhance other cognitive, psychological, and affective features in children with DCD [9]. There have been several movement-based approaches regarding the design of motor skill interventions for developmental coordination disorder, which are commonly classified into two broad categories: those that use movement to target underlying performance issues, commonly determined as process-oriented approaches, and those that utilize activity to address the performance itself, commonly determined as task-oriented approaches [5]. In a study of the efficacy of motor interventions for children with developmental coordination disorder [10,11], the positive impacts of the intervention on the motor performance of children with developmental coordination disorder were observed.

An analysis of the systematic reviews by Saidmamatov et al. [11] and Rodrigues et al. [12] reveals that many interventions have been given to children with DCD to develop their motor skills, and almost all of them have had a positive impact on children with DCD. The given intervention period by researchers typically ranged from 2 weeks [13,14] to 32 weeks [15].

However, if we focus on Yu’s [9] systematic review article, when the effectiveness of interventions that positively affected DCD children was examined after a certain time, some children’s motor abilities, which had improved as a result of the intervention, had regressed to their earlier levels. For example, looking at research conducted by Hammond et al. [16], a 1-month intervention program was able to improve the motor skills of DCD children, but after 2.5 months without intervention, the children’s motor skills, tested again with the Bruininks–Oseretsky Test of Motor Proficiency Second Edition (BOT-2) equipment, returned to their previous state. Similar results were also observed in the studies of Sugden et al. [17] and Wood et al. [18]. However, in other studies, a slightly different result was observed; that is, the results in the post-test were preserved in the follow-up moment [19,20,21,22,23,24]. In a small number of research studies, the motor skills of children with DCD improved from the post-test to the retention test [25]. In order to verify the effectiveness of the given interventions, most of the follow-up tests were re-examined around 1 to 5 months after the end of the intervention program [19,20,21,22]. In the study of Coetzee [25], for example, the effectiveness of the intervention was re-examined two years after the post-test, and it was observed that the overall results of the MABC decreased over time. Based on the above data, we can say that the development of motor skills in children with DCD can be achieved through interventions given over a period of time. However, will the results achieved through the interventions given during a given period remain the same or decrease, and how will the passage of time affect children with DCD? The present study was an extension of our previous work and determined the sustainability of a given ten-week motor skills training program for children with DCD [26]. We thought that motor skills learned during a certain period of intervention would likely go back to how they were before. 

## 2. Methods

### 2.1. Study Design

The research is designed to evaluate whether or not post-test findings change over time. Therefore, the same children who were assessed at the post-test were reassessed 18 months later to determine the intervention’s sustainability.

### 2.2. Participants

This study is a follow-up to a research project undertaken in 2020 [26]. Children with DCD in four kindergartens in the Khorezm region of Uzbekistan were selected to take part in the project. Despite the a priori power calculation that had indicated that a sample size of 30 would provide a power of 0.9 to detect a 5-point difference in the MABC-2 total standard score (mean = 17 and standard deviation = 5) [27], from 63 children measured, only 24 children were eligible and participated in the study. The DCD children were between 4–6 years old (5.25 ± 0.13 years) and participated in the intervention group (IG: 17 children; 10 boys) and control group (CG: 7 children; 4 boys) in 2020. The pre-test and post-test have been previously published [26]. In 2022, they were again approached for participation in this retention test. Of this group, all 24 children were available for this retention test. The composition of these research parts is displayed in Figure 1.

The children were eligible to attend this study if they had not indicated any neurological or physical impairment.

### 2.3. Measuring Instruments

To see how the training program affected both groups, the MABC-2 test was given according to the MABC-2 manual [28]. Details on how the test was performed are given elsewhere [29].

For age band one, sufficient evidence of validity and reliability has been revealed [30,31]. Smits-Engelsman et al. [31] studied the psychometric properties and the confidence of the MABC-2 in 50 typical children (3 years of age) and concluded, based on the test-retest, that even for three-year-olds, the test provides a high degree of confidence, making it sensitive to detecting individual changes. Moreover, Ellinoudis et al. [32], using band 1 on 183 Greek children, advised that the MABC-2 could be a valid and reliable tool to evaluate children aged 3 to 5 years of age with movement difficulties. Henderson, Sugden, and Barnett [33] recommended the following cutoff points from the test manual: ≤5% atypical motor performance, which indicates DCD; 6th to 15th percentile, which means risk of developmental coordination disorder (r-DCD); and any percentile higher than 16%, which demonstrates typical development (TD). The pre-test–post-test was followed by a retention test in which each child was evaluated by the same person and recorded on video.

### 2.4. Intervention

Full details of this intervention are given in the article published by Saidmamatov et al. [26]. Children engaged in daily occupational activities until they were re-tested 18 months later.

### 2.5. Procedure

According to the Declaration of Helsinki, the research was approved by the Ethics Committee of Urgench State University (Code 12356). A meeting was organized with the principals of the respective schools, and the purpose and protocol of the study were explained to them. Written informed consent was obtained from the parents of each child before he/she was allowed to participate in the research. The purpose and procedures of the study were explained to each child before the beginning of the study. These children were evaluated regarding their DCD status.

### 2.6. Data Analysis

SPSS 26.0 was utilized to carry out the statistical analysis, and the statistical significance was evaluated using an alpha level of 0.05. The dependent variables included abilities in the domains of manual dexterity, aiming and catching, balance, and total MABC-2 scores, and for descriptive purposes, data were analyzed using means (M) and standard deviations (SD) and median. Preliminary analyses were performed, including gender as a factor, but no significant effect was found in any of the variables analyzed; therefore, data from both males and females were pooled. The Mann–Whitney was used to test group (IG and CG) differences within the 3 moments. Next, a repeated measures ANOVA with adjustments for sphericity and Bonferroni corrections was utilized to identify whether the children’s scores had changed significantly from baseline and whether any improvements following intervention were maintained after an 18-month retention test within the groups. Furthermore, an estimate of the effect size, by partial eta squared (ηp2), was calculated for each dependent variable. Cohen’s [34] guidelines say that an effect of 0.0099 is small, 0.0588 is medium, and 0.1379 is a large effect. 

## 3. Results

### 3.1. Program Effect

In the retention test, only 6 children of the IG maintained their MABC percentage above 16%. Children in the CG performed almost identically on all three moments (pre-test, post-intervention, and retention test) (Table 1).

### 3.2. Group Effect

Table 2 shows that there were no statistically significant differences between the two groups in all MABC-2 domains measured during the pre-test. However, statistically significant differences were evident in all the MABC-2 domains between the two groups post-intervention after 10 weeks of intervention. Finally, in the retention test, again, no statistically significant differences were found between the two groups on all the MABC-2 domains.

### 3.3. Time Effect

#### 3.3.1. Manual Dexterity

Concerning MD, results show an effect of time (F (2, 21) = 6.514, *p* = 0.006, ηp2 = 0.383) and an interaction between time and group (F (2, 21) = 12.363, *p* < 0.001, ηp2 = 0.541), as can be seen in Figure 2 below.

A follow-up analysis revealed that the IG retained their performance after the retention test since there were no noticeable differences between the post-intervention period and the retention test (*p* > 0.050, post-intervention: 7.76 ± 2.25; retention test: 7.35 ± 3.44).

#### 3.3.2. Aiming and Catching

As seen in Figure 3, the results revealed an effect of time (F (2, 21) = 5.212, *p* = 0.015, ηp2 = 0.332) and an interaction between time and group (F (2, 21) = 2.897, *p* = 0.077, ηp2 = 0.216).

A decline was evident (post-intervention: 13.17 ± 2.24; retention test: 9.82 ± 3.97), but change over time needed to be analyzed within each group separately. In the IG, an improvement was found from the pre- to post-intervention, and a decline was seen from the post-intervention period to the retention test (*p* < 0.050) back to the initial scores. The CG maintained their performance (*p* > 0.050).

#### 3.3.3. Balance

The findings have demonstrated an effect of time (F (2, 21) = 1.149, *p* = 0.336, ηp2 = 0.099), and there was an interaction between time and group (F (2, 21) = 6.673, *p* = 0.006, ηp2 = 0.389), as can be seen in Figure 4.

The effect of time revealed a decline from the post-intervention period (11.11 ± 1.96) to the retention test (9.29 ± 2.49). Multiple analyses revealed that the IG declined in performance from post-intervention to the retention test evaluation back to initial scores (*p* < 0.050). The CG maintained their performance (*p* > 0.050).

#### 3.3.4. Total Test Score MABC-2

As seen in Figure 5, Total Score results have indicated an effect of time (F (2, 21) = 15.145, *p* < 0.001, ηp2 = 0.591) and an interaction between time and group (F (2, 21) = 23.108, *p* < 0.001, ηp2 = 0.688).

Concerning Total Score, although there was a significant decline from the post-intervention period to the retention test (*p* < 0.050, post-intervention: 10.64 ± 1.53; retention-test: 8.47 ± 3.98), children in the IG were better than in the pre-test (*p* < 0.050). The CG maintained performance (*p* > 0.050). The retention test results for the IG revealed a decrease in the MABC-2 overall score. Even though this was the case, the retention test results were better than those of the pre-test.

## 4. Discussion

This research sought to determine whether or not post-test results were altered over time. Consequently, the same children who were evaluated at the post-test were re-evaluated 18 months later to establish whether the allocated intervention was stable in the motor skills training program. Partly similar subjects were also investigated by Sugden et al. [17], Wood et al. [18], Fong et al. [19], Capistran et al. [24], and Coetzee et al. [25], who assessed the sustainability of allocated intervention for children with DCD. Our findings revealed disparities between the intervention group and control group in all the motor competence measures in the three assessment parts, as in other studies [18,19]. Importantly, our study found that the mean scores of the intervention groups’ MABC-2 total score, manual dexterity, aiming, catching, and balance skills were considerably decreased after completion of the motor skill training program. This result partially supports the results of Sugden et al. [17] and Wood et al. [18], and based on the results of these studies, the effectiveness of interventions for children with DCD may decline over time after the end of the intervention. Similar to Wood et al. [18]’s findings, the results of the children’s test dropped significantly from the post-intervention test to the retention test; nonetheless, these results were still superior to the pre-test results, demonstrating that the impact of the intervention is still present in the retention test, showing a more long-term effect. The findings of our research, on the other hand, contradict the conclusions of previous studies [19,20,21,22,23,24]. Nevertheless, this study differed from others [17,18] because their retention tests were conducted after no more than 8 months, showing a short-term effect.

The research also shows that children with DCD may improve their aiming and catching abilities without intervention. This conclusion was also supported by other studies [17,18]. It is important to note, however, that the rate of development in aiming and catching abilities was greater in the intervention group than in the control group in the post-test and retention tests.

Another finding of the research was that following the intervention period, all children with DCD in the intervention group scored over 16% on the MABC-2 [33,34], indicating that their motor abilities improved as a result of the intervention. However, during the retention test, 18 months after the intervention, only six of seventeen children sustained beneficial improvements.

According to these studies, the sustainability of the allocated intervention may not stay long. Therefore, interventions should be given to DCD children in a systematic and continuous manner, and frequent screening is recommended for children with DCD.

## 5. Conclusions

Concerning Total Score, although there was a significant decline from post-intervention to retention test, children in the IG performed better than in the pre-test. The CG maintained performance. The retention test results for the IG revealed a decrease in the MABC-2 overall score. Even though this was the case, the retention test results were better than those of the pre-test. Future research should focus more on implementing intervention programs and evaluating their effects through post-tests and retention tests. Only in this way can we obtain more robust conclusions about the motor behavior changes of children with DCD after their participation in these programs, as well as observe changes in their behavior after a period without intervention. Thus, the reliability of the research will be much higher.

## Figures and Tables

**Figure 1 sports-10-00164-f001:**
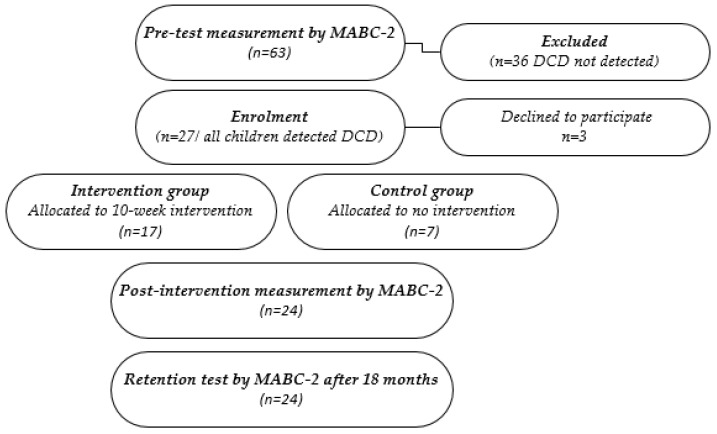
Composition of these research parts of the study.

**Figure 2 sports-10-00164-f002:**
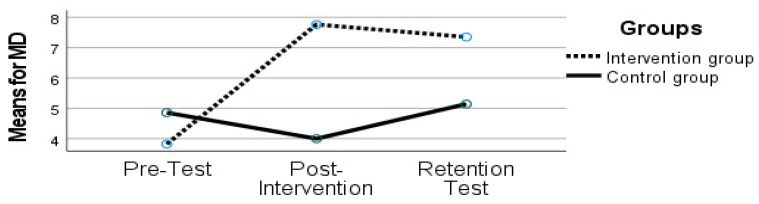
Manual dexterity scores on the Movement Assessment Battery for Children (MABC-2) for both groups at time 1 (pre-test), time 2 (post-intervention), and time 3 (retention test).

**Figure 3 sports-10-00164-f003:**
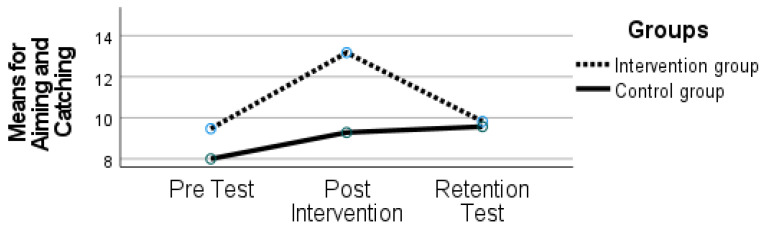
Aiming and Catching scores on the Movement Assessment Battery for Children (MABC-2) for both groups at time 1 (pre-test), time 2 (post-intervention), and time 3 (retention test).

**Figure 4 sports-10-00164-f004:**
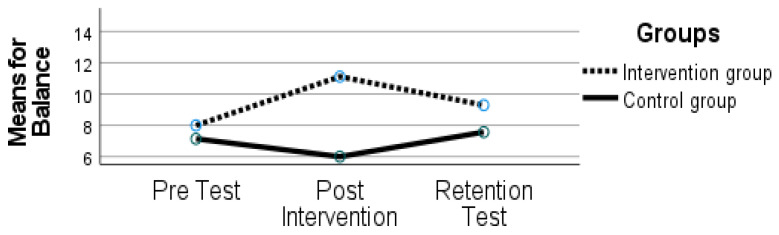
Balance scores on the Movement Assessment Battery for Children (MABC-2) for both groups at time 1 (pre-test), time 2 (post-intervention), and time 3 (retention test).

**Figure 5 sports-10-00164-f005:**
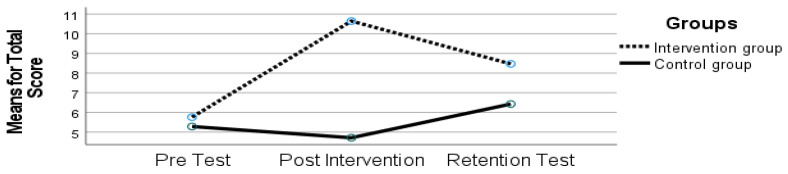
Total test scores on the Movement Assessment Battery for Children (MABC-2) for both groups at time 1 (pre-test), time 2 (post-intervention), and time 3 (retention test).

**Table 1 sports-10-00164-t001:** Number of children scoring in each MABC-2 percentile band at pre-test, post-intervention, and retention test after 18 months.

Motor Difficulty Category	Intervention Group (*n* = 17)			Control Group (*n* = 7)		
	Pre	Post	Retention	Pre	Post	Retention
No motor difficulty (MABC-2 > 16th percentile)	0	17	6	0	1	1
At risk of DCD (from 6th–16th percentile)	11	0	8	3	1	2
Probable DCD (MABC-2 ≤ 5th percentile)	6	0	3	4	5	4

**Table 2 sports-10-00164-t002:** Intervention and control groups. Results from MABC-2 domains at pre-test, post-intervention, and retention test after 18 months.

Variables	Intervention Group	Control Group	*p*
	Mean ± SD (Median; Range)	Mean ± SD (Median; Range)	
Pre-test			
Total Score	5.76 ± 1.39 (6; 3–7)	5.29 ± 1.11 (5; 4–7)	0.294
Manual Dexterity	3.82 ± 1.18 (4; 2–6)	4.86 ± 1.86 (6; 2–7)	0.171
Aiming and Catching	9.47 ± 2.80 (10; 5–15)	8.00 ± 2.70 (8; 3–12)	0.277
Balance	8.00 ± 2.34 (8; 4–12)	7.14 ± 2.19 (7; 4–11)	0.498
Post-intervention			
Total Score	10.64 ± 1.53 (11; 8–14)	4.71 ± 1.88 (4; 2–8)	**<0.001**
Manual Dexterity	7.76 ± 2.25 (9; 4–12)	4.00 ± 1.73 (4; 2–6)	**0.002**
Aiming and Catching	13.17 ± 2.24 (13; 8–17)	9.28 ± 3.35 (8; 6–15)	**0.017**
Balance	11.11 ± 1.96 (11; 9–16)	6.00 ± 1.63 (6; 4–8)	**<0.001**
Retention-test			
Total Score	8.47 ± 3.98 (7; 3–15)	6.43 ± 3.40 (5; 4–14)	0.075
Manual Dexterity	7.35 ± 3.44 (6; 3–13)	5.14 ± 2.19 (4; 4–10)	0.108
Aiming and Catching	9.82 ± 3.97 (9; 4–18)	9.57 ± 4.11 (8; 5–17)	0.822
Balance	9.29 ± 2.49 (9; 6–15)	7.57 ± 1.71 (7; 6–11)	0.076

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
