# Peer review of "Sustainability of a Given Ten-Week Motor Skills Training Program for Children with Developmental Coordination Disorder"

_sports, 2022, doi:10.3390/sports10110164_

Round 1
Reviewer 1 Report
Comments to the Author
The manuscript titled "Sustainability of a given ten-week motor skills training program for children with developmental coordination disorder" analyzes the sustainability of a given ten-week motor skills training program for children with developmental coordination disorder (DCD). However, there are several points that require further clarity;
1- Page 1, Line 20: revise the ‘’per-formance’’ as ‘’performance’’
2- Page 1, Line 22: revise the ‘’inter-vention’’ as ‘’intervention’’
3- Page 1, Line 24: remove the numbers in keywords
4- Page 2, Lines 90-91: Express the sentence more clearly and make sure your study design is cross-over. I think, your study design is not a ''cross-over design''.
5- Page 2, Line 94: Provide anthropometric data in a table.
6- Page 3, Line 102: In Figure 1, you show that the control group consists of 17 children, but I see it is written as 7 children in the ''Participants'' and ''Abstract'' sections. Please explain.!
7- Page 3, Lines 109-123: We need details on how the test was performed rather than on age groups or validations of the test. Please give more detail.!
8- Page 4, Lines 136-150: Sample size analysis is not specified, power analysis is not done.
9- Page 5, Lines 183-184: If you used Mann-Whitney U for analysis, then you have to need median and min-max, not mean and SD. Please give the median and min-max values in table 4
10- Page 5, Lines 183-184: I wonder why you didn't compare by gender. This is a serious lack. Please create a table for that.
11- Pages 6 and 7: Show the differences between times (pre, post, retention) for both groups (control and intervention) in figures 2, 3, 4, and 5 (or just in table 4). I mean which times are similar to one another or which times are different from another? you can use letters or symbols for this.
12- Pages 7, Lines 243 and 246: move numeric values to the results section.
GENERAL COMMENTS:
1. The manuscript requires language improvement.
2. The topic is important but especially the discussion section should be improved
significantly. Literature review is nonadequacy.
3. Abstract should be re-edited after changes made in the article.
Author Response
Thank you for your careful review of our article. Your comments helped to improve the quality of our article.

Reviewer 2 Report
Thank you very much for the opportunity to review this manuscript. The topic of the paper is interesting and fits the scope of the journal. The text is relatively well written and composed. The major limitation of this study is the small sample size of control group, only 7 children. Also, I have many minor comments that I believe that help to improve the paper.
Introduction
Lines 65-67. Please write the following sentence more clearly: “we can see that some children's motor skills improved by the intervention returned to their previous state”.
Line 75. Follow up test, do you mean retention test?
Lines 84-87. Please add the reference of your previous work.
Line 95. Please add the reference of your previous work.
Line 97. Please refer the mean and SD of age.
Results
Table 4. Please check and replace commas with dots.
Discussion
The section of discussion is very poor. Please analyze more the studies that reported. For example in the lines 258-261 analyze more these studies. Also, in the line 262 analyze more these studies.
Author Response
Thank you for your careful review of our article. Your comments helped to improve the quality of our article

Round 2
Reviewer 1 Report
General comments:
This study contains too many similarities with the previous article titled "A Ten-Week Motor Skills Training Program Increases Motor Competence in Children with Developmental Coordination Disorder" by the authors (https://doi.org/10.3390/children8121147). As far as I can see, the only difference between the current study and the previous study is that it contains retention data. However, the authors did not clearly state that they used the data from their previous papers in this study.
In addition, I found that the introduction, method, result, and reference sections in the study are largely similar to the previous work (the sentence structures are different, but even the reference order is the same). Finally, there were noticeable differences in the number of participants between the previous study and the current study, and the reason for this difference was unclear; however, the authors revised it, saying they made a mistake (in revision 2). I think these revisions are insufficient and present significant originality issues for the study.
Best regards
Author Response
Dear reviewer
Thank you for your comments. Please find my attached responses below.

Reviewer 2 Report
I would like to thank the authors for considering my comments and changing the manuscript accordingly. In my opinion the manuscript has been improved by the more focused discussion. The text of discussion is very poor. Please analyse more the discussion.Author Response
Dear reviewer
Thank you for your comments. Please find my attached responses below.
